# Smartphone-based point-of-care anemia screening in rural Bihar in India

Verena Haggenmüller[1], Lisa Bogler [2✉], Ann-Charline Weber[2], Abhijeet Kumar[2], Till Bärnighausen[1], Ina Danquah[1] & Sebastian Vollmer [2]

## Abstract

**Background** The high prevalence of anemia in resource-constrained settings calls for easy-to-use, inexpensive screening tools. The Sanguina Smartphone App, an innovative tool for non-invasive hemoglobin estimation via color-sensitive, algorithm-based analysis of fingernail bed images, was validated in the United States. This study evaluates the performance of the App in a population with different socio-economic, ethnic, demographic and cultural composition in rural Bihar, India.

**Methods** For 272 mainly adult patients of a private health centre, hemoglobin measurement with the App is compared with the gold standard laboratory blood analysis. For a second sample of 179 children attending pre-schools, hemoglobin measurement with the App is compared to the results of the HemoCue Hb 301, a point-of-care device using a small blood sample, serving as the reference standard for field-based settings.

**Results** The App reaches ±4.43 g/dl accuracy and 0.38 g/dl bias of comparator values in the clinic-based sample, and ±3.54 g/dl and 1.30 g/dl, respectively in the pre-school sample. After retraining the algorithm with the collected data, the validity of the upgraded version is retested showing an improved performance (accuracy of ±2.25 g/dl, bias of 0.25 g/dl), corresponding to the results of the original validation study from the United States.

**Conclusions** The initial version of the App does not achieve the accuracy needed for diagnosis or screening. After retraining the algorithm, it achieves an accuracy sufficient for screening. The improved version with the potential for further adaptions is a promising easy-to-use, inexpensive screening tool for anemia in resource-constrained point-of-care settings.

## Plain language summary

Hemoglobin is a protein in the blood that is required to move oxygen around the body. Anemia is a condition that occurs when levels of hemoglobin are too low. Anemia can lead to serious problems with the heart and lungs. This study assesses the accuracy of using a Smartphone App to measure hemoglobin levels by taking photos of a person's nails in rural Bihar, India. Results from the App are compared with those obtained by measuring hemoglobin in blood samples. Initially, measurements of hemoglobin obtained using the App were inaccurate, however when the App was modified using data from some of the people in India, the App was more accurate. This study shows that the App can and should be adapted for use in different populations and can enable anemia to be diagnosed outside of hospitals and other healthcare settings.

[1] Heidelberg Institute of Global Health, Heidelberg University Hospital, Im Neuenheimer Feld 130.3, 69120 Heidelberg, Germany. [2] Centre for Modern Indian Studies (CeMIS), University of Goettingen, Waldweg 26, 37073 Göttingen, Germany. ✉email: lisa.bogler@cemis.uni-goettingen.de

Anemia still constitutes a serious global health problem. It is defined as low blood hemoglobin concentration (Hb), the main protein in erythrocytes. Its main function is to take up oxygen from the lungs and to deliver it to the tissues, as well as to transport carbon dioxide from the peripheral tissues back to the lungs. As of 2019, the prevalence of anemia in all age groups is estimated to be 22.8% globally[1] and low- and middle-income countries (LMIC) account for 89% of all anemia-related disabilities[2].

Anemia has multiple adverse effects on human health and a substantial impact on socioeconomic development, in both, the short and long term[3,4]. Moreover, anemia is associated with negative effects on children's cognitive, motor and social-emotional development, which adversely affects school performance[5]. Anemia during pregnancy is linked to poor pregnancy and birth outcomes, including an increased risk of maternal and neonatal mortality[6–9]. Furthermore, higher susceptibility to infections as well as reduced work productivity have been found for all age groups. All these effects also translate into reduced income of affected people[10,11]. In 2019, anemia was responsible for 58.6 million years of life with disability[1].

Screening measures are a first step in defeating anemia. However, especially in LMICs, progress in this regard has been too slow and is insufficient to date[12]. This is partly due to the lack of adequate tools for determining Hb outside clinical settings.

A variety of methods and devices are available for Hb determination. The current gold standard is performed via complete blood count (CBC), using automated hematology analyzers. Its material-, personnel-, cost- and time-intensive requirements, however, impair its use, especially in peripheral health centres. Point-of-care (POC) methods can be divided into invasive, e.g. HemoCue or WHO color scale, and non-invasive techniques, e.g. NBM-200™ (OrSense) or Masimo devices[13–16]. Invasive procedures expose medical staff to potentially infectious material and possibly cause pain and discomfort among patients, especially young children. This is where non-invasive techniques offer key advantages. The Sanguina Smartphone App has been proposed as an innovative tool, which showed promising results in a first study presented by Mannino and colleagues[17]. The authors compared Hb measurements via App with CBC as reference test in 100 patients in the United States and reported an accuracy of ±2.4 g/dl (95% limits of agreement) and a mean difference of 0.2 g/dl from the comparator, highlighting its potential for anemia screening purposes.

The App has so far only been tested in a controlled laboratory setting in the United States under optimal test conditions[17] and during health screenings of refugees arriving in the United States[18]. Thus, there is uncertainty about the performance of the App in the field and in other settings with different socio-economic, ethnic, demographic and cultural composition of the population. This research aims to evaluate the two key indicators of performance, validity and reliability, of the Sanguina App in comparison to established reference techniques in two different samples within a low-resource environment in rural Bihar, India. The App initially has a relatively high measurement inaccuracy. Retraining the algorithm with collected data leads to a considerable improvement in performance. This improvement shows the substantial learning capacity of the App and its ability to be adapted to a different population.

## Methods

**Study setting and participants.** The study was conducted in the district of Madhepura in the state of Bihar, India, between December 2019 and January 2020. This district is predominantly rural and one of the poorest in India. The study encompassed two different samples. First, a clinic-based sample comprised 275 outpatient attendees in a community-based clinic (*Dr. Sitaram Yadav Clinic*) in the city of Madhepura. Attendees were either patients who were referred for a hematological investigation or their accompanying family members. Second, a pre-school sample comprised 197 children aged 2 to 7 years in five pre-schools, so-called Anganwadi Centres (AWC), in remote villages in the district of Madhepura. All children who were present at the time of visit were measured.

**Estimating Hb using the Sanguina Smartphone App.** In both samples, the Sanguina Smartphone App was used as the method to be evaluated and compared with the respective reference technique. Its functionality is based on an extremely color-sensitive analysis algorithm which evaluates the image data of photos taken from fingernail beds[17]. After cleaning the nails and removing nail polish, the participants were asked to bend their fingers inwards to generate a hand and finger positioning recommended by the App's developers. After having taken the picture, the enumerator manually selected the regions of interest (ROI) by placing a box in the center of the nail bed of each finger, except the thumb. In doing so, the enumerator could choose regions in a way that irregularities such as discoloration or dirt were excluded from analysis. Within seconds the calculated Hb value was displayed on the screen. In the clinic-based sample, a total of 5 photos per participant were captured, either of the right or left hand, in some cases both. In the pre-school sample, between 1 and 4 photos were taken per child, either of the right or left hand, depending on the child's willingness to cooperate and ability to keep the required finger position stable. Pictures were obtained with the same smartphone model (iPhone 5s, the model for which the App was developed at the time of data collection).

Fingernail discolorations, e.g. due to medical conditions, injuries, henna or nail polish were prevalent and led to measurement inaccuracies of the extremely color-sensitive algorithm of the App. Therefore, the photos were divided into four categories (clinic-based sample, $n = 259$) (Fig. 1, panel a) and three categories (pre-school sample, $n = 157$) (Fig. 1, panel b). This classification was done by the investigators by means of visual inspection, taking into account the respective criteria of the different categories defined a priori. Statistical analysis was performed for both, the entire study population as well as subgroups of nail categories.

**Blood sampling and laboratory measurements.** In the clinic-based sample, a random venous blood sample for CBC evaluation, run by the *Aspen Mindray BC-5000* autoanalyzer (Mindray, China), was taken to determine Hb reference values. This autoanalyzer was the best possible local standard, which has been shown to have an acceptable overall performance (coefficient of variation of Hb: 0.58–1.06)[19]. In the pre-school sample, capillary Hb estimation was conducted using the HemoCue Hb 301 device (HemoCue AB, Sweden), a handheld photometer which is commonly used in field-based settings[20] and is considered as reference standard in a setting where no lab access is available. The performance of the HemoCue was investigated in a subgroup of the clinic-based sample (HemoCue vs. laboratory reference method, $n = 106$) and compared with literature results prior to data collection in pre-schools.

The WHO-recommended age- and sex-specific Hb cut-offs were used to define anemia and classify severity, based on the respective Hb reference values of the two study samples[21]. 6-59 months old children and pregnant women ≥15 years of age were classified as having anemia, when Hb <11 g/dl and 5-11 years old children when Hb <11.5 g/dl. For teens between 12 and 14 years of age and for non-pregnant women (≥15 years), the cut-off indicating anemia was <12 g/dl, for men ≥15 years of age it was

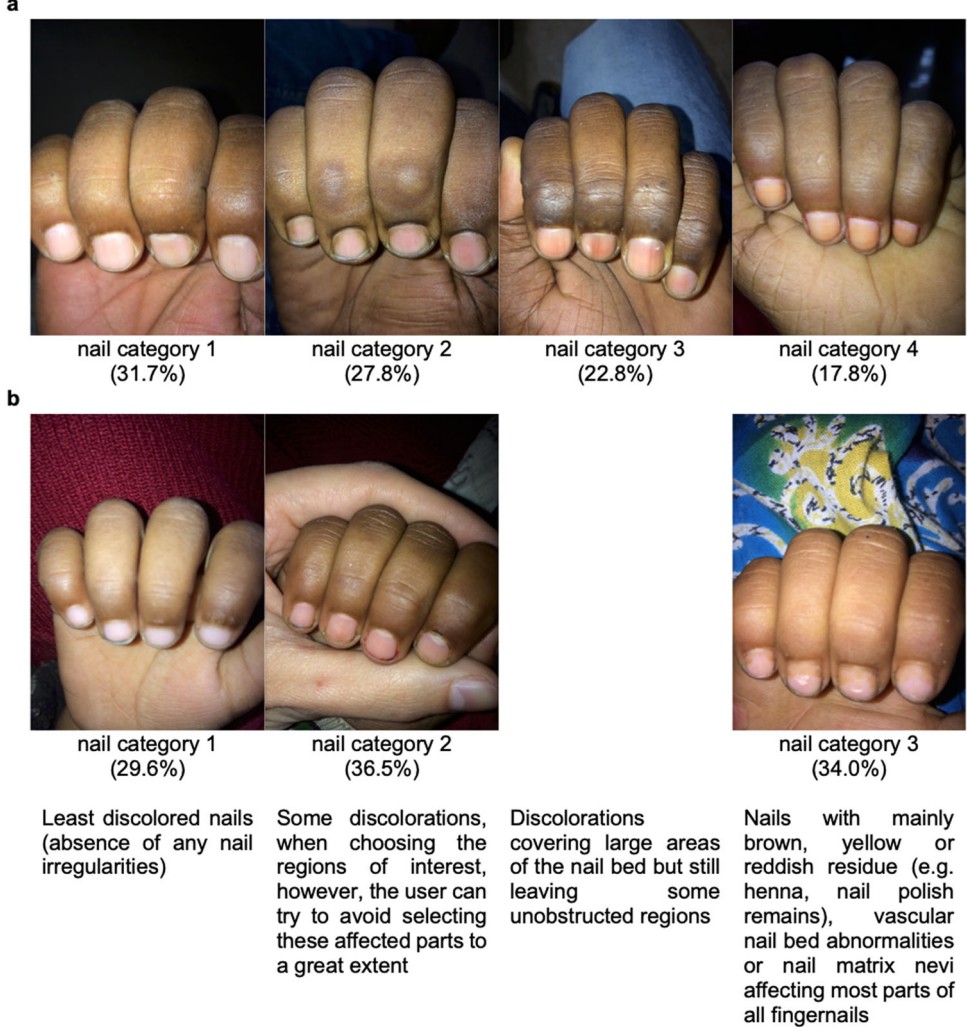

**Fig. 1 Definition of nail categories.** One representative picture of each nail category in **a** the clinic-based and **b** the pre-school sample with respective criteria and prevalence.

<13 g/dl. In case of anemia, the patient was referred to the responsible physician in the clinic-based sample and a definitive confirmatory testing via CBC was strongly recommended for the affected children in the pre-school sample. Hb assay among participants was performed by qualified medical personnel with all measurements being taken in the same session with no treatment having been administered in between. The same set of equipment was used for all patients.

**Assessment of participants' characteristics**. Additionally, age, sex, and diet requirements (vegetarian or non-vegetarian) were recorded in the clinic-based sample. Demographic characteristics, including age (in years, reported by parents) and sex, as well as anthropometric measurements of height, weight and mid-upper arm circumference (MUAC) were obtained from children of the pre-school sample. Explanations and results of the anthropometric data are provided in the Supplementary Methods (Section 1.1).

**Statistical analysis**. The descriptive characteristics of the study population are presented as means ± standard deviations (SD) and ranges for normally distributed continuous variables or as medians and interquartile ranges (IQR) for non-normally distributed variables, and as counts and proportions for categorical variables. Shapiro-Wilk tests were done to assess normality.

For the assessment of the validity of the App, pairs of Hb readings were compared. In the clinic-based sample, the Sanguina Smartphone App (using the first Hb App value) was compared against the automated hematology analyzer. In the pre-school sample, the Sanguina Smartphone App (using the first Hb App value) was evaluated against the HemoCue Hb 301. A paired *t*-test was used to determine statistical significance of difference between the mean Hb values of the respective comparison pairs. The Bland-Altman technique was applied to the study data, and the bias and 95% limits of agreement (LoA), determined as bias ±1.96 x SD, were calculated[22,23]. Bias was defined as the mean difference between the two measurements (App – reference method) and was expressed in g/dl. Bland-Altman plots with the difference (y-axis) plotted against the average of the paired values (x-axis) were generated for graphical visualization of the agreement between the methods across the entire range of measured Hb[24,25].

In order to evaluate the accuracy from a clinical perspective, clinically acceptable limits of ±1 g/dl were defined which correspond to the common accuracy requirements for anemia diagnosis. Anemia screening, however, especially in POC settings, is subject to less stringent accuracy demands. A calculation of the proportion that fell within a range of ±1 g/dl to ±2.4 g/dl (diagnostic accuracy reported by Mannino and colleagues as representative range for POC settings[17]) from the reference value

**Table 1 Patients' characteristics including age, vegetarian status and anemia prevalence in the clinic-based sample.**

| Characteristics | Total | | Men | | Women | |
|---|---|---|---|---|---|---|
| | *n* | % | *n* | % | *n* | % |
| | **272** | **100** | **160** | **58.8** | **112** | **41.2** |
| Age (years, median, IQR) | 33.5 (22; 49) | | 33 (22; 46) | | 35 (22; 50) | |
| Vegetarian | 73 | 26.8 | 37 | 23.1 | 36 | 32.1 |
| Anemia[a] | 123 | 45.2 | 51 | 31.9 | 72 | 64.3 |
| Mild anemia | 80 | 29.4 | 43 | 26.9 | 37 | 33.0 |
| Moderate anemia | 38 | 14.0 | 6 | 3.8 | 32 | 28.6 |
| Severe anemia | 5 | 1.8 | 2 | 1.3 | 3 | 2.7 |

*IQR* inter-quartile range.
[a]Hemoglobin (g/dl) cut-offs to define anemia according to[21], based on blood test.

**Table 2 Children's characteristics including age and anemia prevalence in the pre-school sample.**

| Characteristics | Total | | Boys | | Girls | |
|---|---|---|---|---|---|---|
| | *n* | % | *n* | % | *n* | % |
| | **179** | **100** | **81** | **45.3** | **98** | **54.8** |
| Age (years, mean ± SD, range) | 4.5 ± 1.2 (2–7) | | 4.5 ± 1.1 (2–7) | | 4.5 ± 1.3 (2–7) | |
| Anemia[a] | 43 | 24.0 | 23 | 28.4 | 20 | 20.4 |
| Mild anemia | 26 | 14.5 | 17 | 21.0 | 9 | 9.2 |
| Moderate anemia | 16 | 8.9 | 6 | 7.4 | 10 | 10.2 |
| Severe anemia | 1 | 0.6 | 0 | 0.0 | 1 | 1.0 |

*SD* standard deviation.
[a]Hemoglobin (g/dl) cut-offs to define anemia according to[21], based on HemoCue measurement.

was performed. Furthermore, we included an assessment of sensitivity and specificity. A receiver-operating characteristic (ROC) analysis was performed, which displays graphically the true positive rate (sensitivity) versus the false positive rate (1 – specificity). The area under the ROC curve (AUC) can take values between 0 and 1 and is a measure of how well a test is able to distinguish between two diagnostic groups, here anemic and non-anemic patients.

The test-retest reliability of the App was investigated by measuring the consistency of repeated Hb App values (5 consecutive photos per subject within the clinic-based sample and 3 within the pre-school sample) taken by a single operator using an intraclass correlation coefficient (ICC).

Image metadata and color data of the photos of those in the clinic-based sample without nail irregularities (nail category 1, see Fig. 1) were used by the App developers to retrain the App as follows. Testing images were randomly selected from the clinic-based sample. The remaining images were used together with the dataset reported in the original study[17] to train the new algorithm. Color data from the images as well as image metadata were correlated with laboratory-determined hemoglobin levels using robust multi-linear regression with a bisquare weighting algorithm (described in detail in ref. [17]). The developers used the resulting correlation coefficients to estimate hemoglobin levels for the testing images, and the resulting estimates were compared with laboratory-determined hemoglobin levels to determine algorithm accuracy. This entire process was repeated multiple times to ensure that the best possible algorithm for the newly expanded dataset was selected. This new algorithm was then incorporated into the App by its developers. This study then assessed the effectiveness of this retraining of the algorithm by a re-analysis of validity of a subset of images of the pre-school sample ($n = 160$).

The relationship between nail category and validity of the App was investigated by applying the same statistical procedures to the subgroup of nails in categories 1 and 2 when compared to the respective reference test. Results of this analysis are described in the Supplementary Methods (Section 1.2).

All statistical analyses were conducted using STATA SE 15. Participants with missing data were excluded from analysis (see Supplementary Figs. 1-3). A *p*-value of ≤0.05 was used to indicate statistical significance.

**Ethical approval and consent to participate**. The study protocol was reviewed by the Ethics Committee of the University of Goettingen, Germany and received a statement of no objection. Prior to enrolment, study procedures and details were explained to the participants and children's parents or legal guardians, respectively, by the study investigators. All participants or their caregivers gave written informed consent.

**Reporting Summary**. Further information on research design is available in the Nature Portfolio Reporting Summary linked to this article.

## Results

**Characteristics of the study population**. Table 1 shows the participants' characteristics of the clinic-based sample ($n = 272$), for the total population and by sex. The median age of the 112 women and 160 men was 33.5 years. 26.8% of the study population were vegetarians. In total, 45.2% of the sample had anemia, with 29.4% having mild anemia, based on the Hb reference values measured by the automated hematology analyzer. Women more often had anemia than men (64.3% vs. 31.9%).

Table 2 shows the characteristics of 81 boys and 98 girls in the pre-school sample ($n = 179$). The children had a mean age of 4.5 ± 1.2 years. The proportion of anemia was 24.0%, based on the Hb reference values measured by the HemoCue photometer. More boys than girls had anemia (28.4% vs. 20.4%). The results of the anthropometric data are presented in Supplementary Table 1.

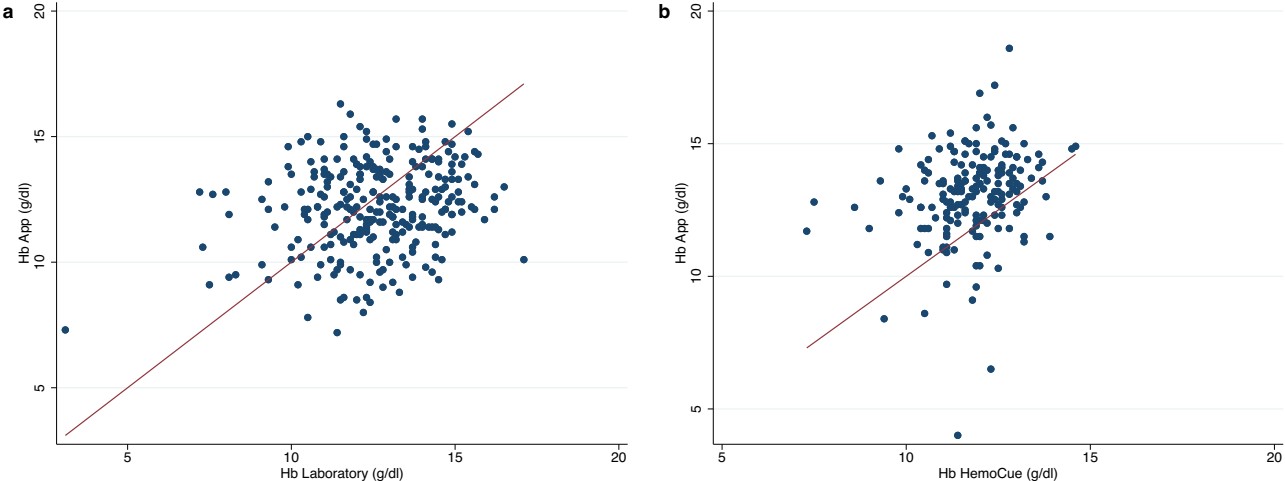

**Fig. 2 Hemoglobin measurements by App and reference method.** Scatterplot showing App hemoglobin (Hb) measurements (y-axis) plotted against **a** the laboratory reference (x-axis) in the clinic-based sample ($n = 272$) and **b** the HemoCue reference in the pre-school sample ($n = 179$). The red diagonal line represents the line of equality where App Hb concentration is equal to the respective reference Hb concentration. Source data for **a** is in Supplementary Data 1; source data for **b** is in Supplementary Data 2.

39.9% of the children were stunted and 2.2% were affected by acute malnutrition, measured as MUAC below 12.5 cm for children younger than 60 months.

**Validity**. Within the clinic-based sample, mean Hb by the laboratory reference method was $12.58 \pm 1.88$ g/dl (Supplementary Table 2). This figure was significantly lower when using the App measurements ($12.20 \pm 1.75$ g/dl, $p = 0.006$). The scatterplot in Fig. 2, panel a reveals a relatively wide scattering along the line of equality and the Pearson correlation coefficient was $r = 0.225$ ($p = 0.0002$). Bland-Altman analysis showed 95% LoA of $-4.81$ to $4.05$ g/dl, and a mean difference of $-0.38$ g/dl, indicating an accuracy of $\pm 4.43$ g/dl (Fig. 3, panel a). The average error magnitude was 1.88 g/dl (Table 3, panel a) and 31.3% of Hb App values lay in the range of $\pm 1$ g/dl deviation from the reference, 67.3% were within $\pm 2.4$ g/dl (Table 4, panel a). A sensitivity of 51.2% and a specificity of 41.6% were found (Table 5, panel a). The positive predictive value (PPV) and negative predictive value (NPV) were 42.0% and 50.8%, respectively. We conducted Bland-Altman analysis separately for men and women as well as individuals with mild or no anemia and those with moderate or severe anemia. Bland-Altman analysis for men showed 95% LoA of $-4.93$ to $2.40$ g/dl, and a mean difference of $-1.26$ g/dl, indicating an accuracy of $\pm 3.67$ g/dl, while the respective numbers for women were 95% LoA of $-3.37$ to $5.14$ g/dl, and a mean difference of $0.89$ g/dl, indicating an accuracy of $\pm 4.26$ g/dl. Bland-Altman analysis for subjects with mild or no anemia showed 95% LoA of $-4.77$ to $3.07$ g/dl, and a mean difference of $-0.85$ g/dl, indicating an accuracy of $\pm 3.92$ g/dl. Similarly, Bland-Altman analysis for subjects with moderate or severe anemia showed 95% LoA of $-1.66$ to $5.88$ g/dl, and a mean difference of $2.11$ g/dl, indicating an accuracy of $\pm 3.77$ g/dl. The ROC analysis was performed separately for men and non-pregnant women to take into account the different cut-offs for anemia diagnosis. ROC analysis for men $\geq 15$ years of age ($n = 155$), who are considered anemic starting at Hb levels $<13$ g/dl, revealed an AUC of 0.65 (Fig. 4, panel a). For non-pregnant women in the same age group ($n = 110$) with a cut-off of $<12$ g/dl for anemia diagnosis, an AUC of 0.50 was identified (Fig. 4, panel b).

In a subgroup of the clinic-based sample ($n = 106$), the HemoCue was tested against the laboratory reference standard prior to data collection in pre-schools. The calculated accuracy of $\pm 1.26$ g/dl in Bland-Altman analysis is consistent with findings

from other studies[26] and considered sufficient for screening requirements in a field-based context.

Within the pre-school sample, a mean Hb concentration of $11.79 \pm 1.11$ g/dl was determined for the HemoCue reference method compared to $13.09 \pm 1.71$ g/dl for App measurements (Supplementary Table 2). The paired $t$-test revealed a statistically significant difference in the mean Hb concentration between the two methods ($t = 9.67$, $p = 0.000$). The Pearson correlation coefficient of $r = 0.235$ ($p = 0.0015$) indicated that there was a relatively weak positive correlation between App and reference values. The rather wide distribution along the equality line in the associated scatterplot (Fig. 2, panel b) also confirmed this observation. The Bland-Altman technique demonstrated 95% LoA from $-2.23$ to $4.84$ g/dl and a mean difference of $1.30$ g/dl (Fig. 3, panel b). This yielded an accuracy of $\pm 3.54$ g/dl (Table 3, panel b). The in-range analysis showed that 33.5% of the Hb App values deviated $\pm 1$ g/dl from the reference value, 74.3% could be assigned to the deviation range of $\pm 2.4$ g/dl (Table 4, panel b). A large discrepancy between high specificity of about 93% and low sensitivity of about 21% was observed (Table 5, panel b). The AUC of the ROC curves for the children $<5$ years of age (Hb cut-off for anemia diagnosis: $<11$ g/dl) and those $\geq 5$ years of age (Hb cut-off: $<11.5$ g/dl) were 0.62 and 0.68, respectively (Fig. 5).

The distribution of Hb measured by the respective method, pairwise comparison of means (reference vs. App) using a paired $t$-test and Pearson correlation coefficient r are shown in Supplementary Table 2.

**Reliability**. Within the clinic-based sample, a total of five consecutive App measurements of the right hand in 185 participants and of the left hand in 80 participants were performed. The differences in mean, SD and 95% CI for the replicates were very similar. According to the ICC for a two-way mixed-effects model, the repeatability observed between the successive measurements was very high. The ICC for the right hand was 0.96 (95% CI: 0.94; 0.96) and the ICC for the left hand was 0.95 (95% CI: 0.94; 0.97).

Within a subgroup of 162 children of the pre-school sample, a total of three consecutive App measurements of the right or left hand were performed. The observations were very similar in terms of mean, SD and 95% CI. The ICC of 0.69 (95% CI: 0.62; 0.75) also implied relatively high test-retest reliability. The lower ICC in the pre-school sample compared to the clinic-based

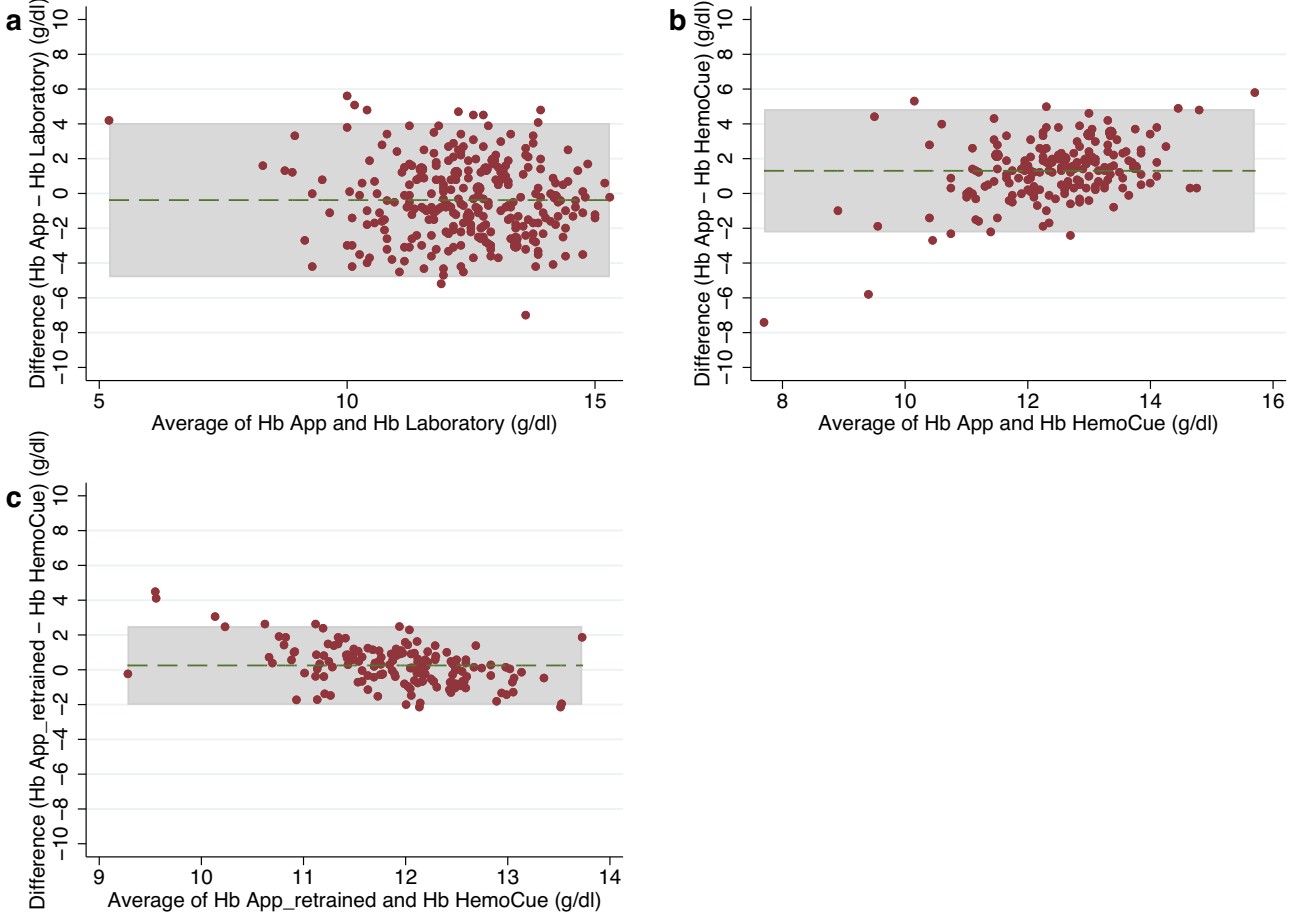

**Fig. 3 Bland-Altman plot.** Bland-Altman plot. The green dashed line represents the mean difference (bias), the gray shaded area represents the 95% limits of agreement (LoA), determined as bias ±1.96 SD; **a** for clinic-based sample, difference (Hb App – Hb Laboratory) on y-axis, average of Hb App and Hb Laboratory on x-axis, bias = −0.38 g/dl, LoA = −4.81, 4.05 (n = 272); **b** for pre-school sample, difference (Hb App – Hb HemoCue) on y-axis, average of Hb App and Hb HemoCue on x-axis, bias = 1.30 g/dl, LoA = −2.23, 4.84 (n = 179); **c** for retrained App, difference (Hb App_retrained – Hb HemoCue) on y-axis, average of Hb App_retrained and Hb HemoCue on x-axis, bias = 0.25 g/dl, LoA = −2.01, 2.50 (n = 160). Source data for **a** is in Supplementary Data 1; source data for **b** and **c** is in Supplementary Data 2.

### Table 3 Validity of Smartphone App.

| | Comparison of methods | n | Average error magnitude | Bias ± SD (95% CI) | 95% LoA | Accuracy |
|---|---|---|---|---|---|---|
| a | Sanguina App vs. hematology analyzer | 272 | 1.88 | −0.38 ± 2.26 (−0.65; −0.11) | −4.81 to 4.05 | ±4.43 |
| | Sanguina App vs. hematology analyzer (nail categories 1 and 2) | 154 | 1.70 | −0.025 ± 2.13 (−0.36; 0.32) | −4.21 to 4.16 | ±4.19 |
| b | Sanguina App vs. HemoCue | 179 | 1.77 | 1.30 ± 1.80 (1.04; 1.57) | −2.23 to 4.84 | ±3.54 |
| | Sanguina App vs. HemoCue (nail categories 1 and 2) | 103 | 1.78 | 1.54 ± 1.53 (1.24; 1.84) | −1.46 to 4.54 | ±3.00 |
| c | Sanguina App vs. HemoCue | 160 | 1.74 | 1.44 ± 1.60 (1.19; 1.69) | −1.69 to 4.58 | ±3.14 |
| | Sanguina App retrained vs. HemoCue | 160 | 0.90 | 0.25 ± 1.15 (0.07; 0.43) | −2.01 to 2.50 | ±2.25 |

Average error magnitude (g/dl), bias ± standard deviation (SD) with 95% Confidence Interval (CI) in parentheses (g/dl), 95% limits of agreement (LoA) (g/dl) and accuracy (g/dl) for the respective comparison of methods; **a** clinic-based sample, **b** pre-school sample, **c** after App retraining.

sample is driven by a few outliers within the series of three values for individual children in the pre-school sample. These were caused by sub-optimal pictures when children did not hold their hand in the optimal position. Supplementary Tables 3 and 4 summarize the results of the reliability analysis.

**Validity after retraining of the App.** The mean of the Hb measurements with the retrained App was 12.02 g/dl compared to 11.77 g/dl determined by HemoCue for the same 160 observations (Supplementary Table 2). Using Bland-Altman method, an accuracy of ±2.25 g/dl (95% LoA: −2.01; 2.50) and a mean

difference of 0.25 g/dl was calculated (Fig. 3, panel c). The average error magnitude was 0.90 g/dl (Table 3, panel c). The sensitivity (20%) showed a slight increase, the specificity remained high at 90%. PPV and NPV stayed almost unchanged at 40% and 77%, respectively (Table 5, panel c).

**Relationship of nail category with validity of the App.** In the clinic-based sample, analysis of validity using a subset of nails in categories 1 and 2 for comparison with the reference revealed 95% LoA of ±4.19 g/dl and bias of −0.025 g/dl (Table 3, panel a and Supplementary Fig. 5, panel a). ROC analysis for nail

**Table 4 Results of in-range-method.**

| | Method of measurement | n | Within ± 1 g/dl | Within ± 2.4 g/dl | Within ± 10% | Within ± 15% | Within ± 20% | Mean divergence | SD |
|---|---|---|---|---|---|---|---|---|---|
| a | Sanguina Smartphone App | 272 | 31.3 | 67.3 | 40.4 | 56.3 | 71.3 | 15.7 | 0.139 |
| | Sanguina Smartphone App (nail categories 1 and 2) | 154 | 39.0 | 73.4 | 47.4 | 63.0 | 77.3 | 14.6 | 0.153 |
| b | Sanguina Smartphone App | 179 | 33.5 | 74.3 | 35.8 | 57.5 | 74.3 | 15.7 | 0.131 |
| | Sanguina Smartphone App (nail categories 1 and 2) | 103 | 31.1 | 73.8 | 35.9 | 57.3 | 71.8 | 16.1 | 0.133 |

In-range-method for App measurements showing the percentage of observations (%) within a defined range of deviation from the reference method (±1 g/dl, ±2.4 g/dl, ±10%, ±15% and ±20%), the mean divergence (%), and standard deviation (SD) of divergence; **a** clinic-based sample, **b** pre-school sample.

**Table 5 Results of sensitivity analysis.**

| | Method of measurement | Sensitivity (%) | Specificity (%) | PPV (%) | NPV (%) | AUC (group 1) | AUC (group 2) |
|---|---|---|---|---|---|---|---|
| a | | | | | | (men, Hb<13 g/dl) | (women, Hb<12 g/dl) |
| | Sanguina Smartphone App | 51.22 | 41.61 | 42.00 | 50.82 | 0.65 | 0.50 |
| | Sanguina Smartphone App (nail categories 1 and 2) | 41.79 | 49.43 | 38.89 | 52.44 | 0.59 | 0.53 |
| b | | | | | | (<5 years, Hb<11 g/dl) | (≥5 years, Hb<11.5 g/dl) |
| | Sanguina Smartphone App | 20.93 | 93.38 | 50.00 | 78.88 | 0.62 | 0.68 |
| | Sanguina Smartphone App (nail categories 1 and 2) | 11.11 | 96.05 | 50.00 | 75.26 | - | - |
| c | | | | | | | |
| | Sanguina Smartphone App | 17.50 | 94.17 | 50.00 | 77.40 | | |
| | Sanguina Smartphone App retrained | 20.00 | 90.00 | 40.00 | 77.14 | | |

Summary of sensitivity, specificity, positive predictive value (PPV), negative predictive value (NPV) (%) and area under the curve (AUC) (receiver-operating characteristic (ROC) analysis) for App measurements; **a** clinic-based sample, **b** pre-school sample, **c** after App retraining.

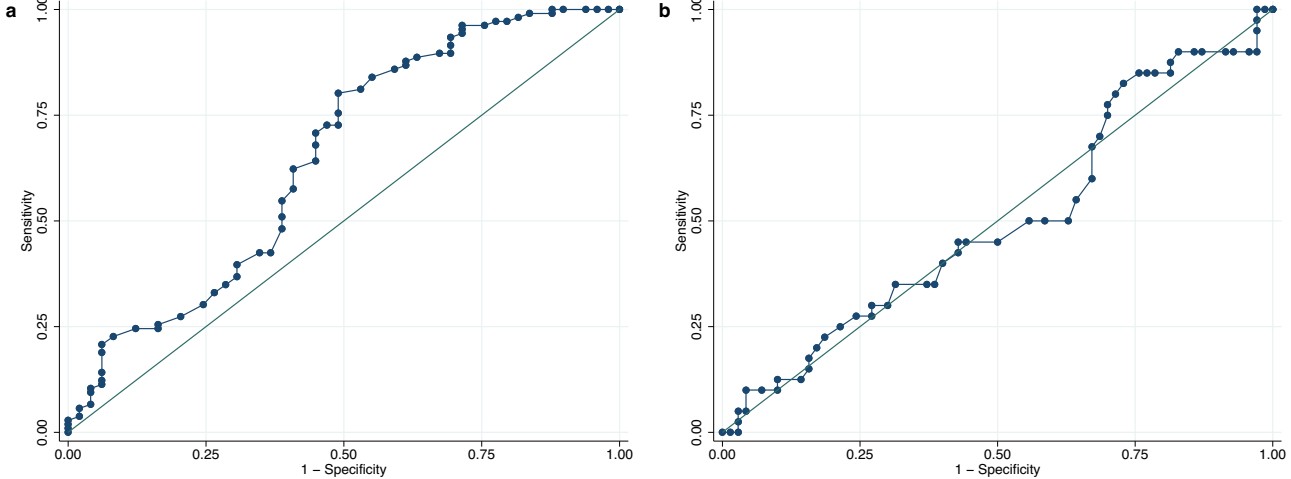

**Fig. 4 Receiver-operating characteristic (ROC) curve for clinic-based sample.** Receiver-operating characteristic (ROC) curve with an area under the curve (AUC) of **a** 0.65 for men (≥15 years of age, n = 155) and **b** 0.50 for women (≥15 years of age, n = 110) in the clinic-based sample. Source data is in Supplementary Data 1.

categories 1 and 2 revealed an AUC of 0.59 for men and 0.53 for women (Supplementary Fig. 6). Within the pre-school sample, Hb values of nail categories 1 and 2 were measured within ±3.00 g/dl and the bias was 1.54 g/dl (Table 3, panel b and Supplementary Fig. 5, panel b).

## Discussion

This study constitutes a performance testing of the Sanguina Smartphone App as a POC device to estimate Hb from fingernail bed images. The App reached ±4.43 g/dl accuracy and 0.38 g/dl bias

of comparator values in a clinic-based sample consisting of 272 study subjects, mainly adults. Comparing men and women, we found a worse performance of the App among women. One possible explanation for the poorer performance among women may be higher variations of Hb in women due to their menstrual cycle[27]. In a pre-school sample of 179 children we found ±3.54 g/dl accuracy and 1.30 g/dl bias.

The findings revealed a considerable degree of measurement inaccuracy of the App. This showed that the App was not suitable for use in a population that was different from the population in which it was calibrated. However, after the developers had

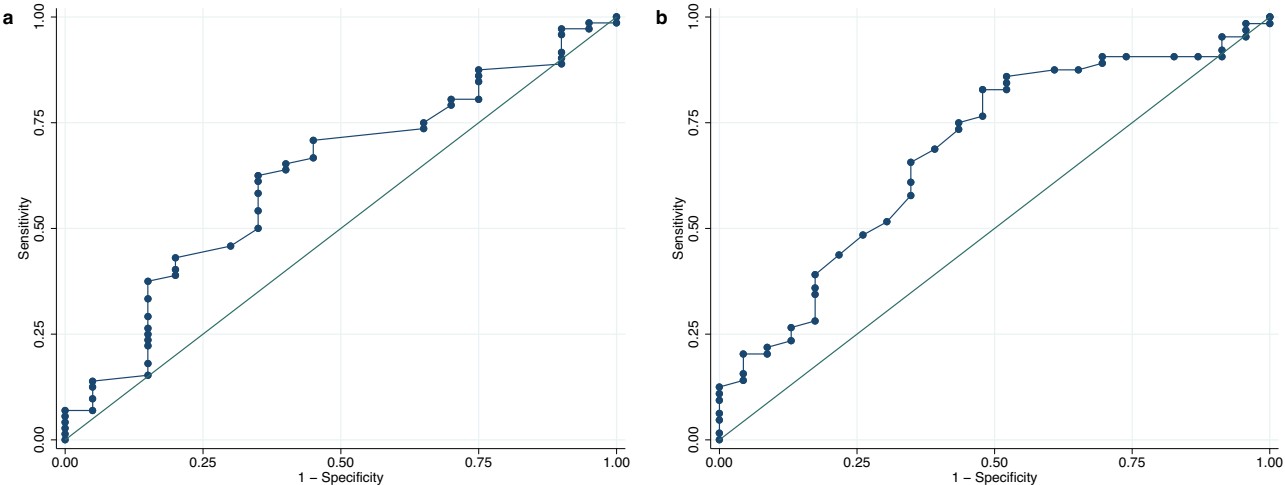

**Fig. 5 Receiver-operating characteristic (ROC) curve for pre-school sample.** Receiver-operating characteristic (ROC) curve with an area under the curve (AUC) of **a** 0.62 for children <5 years of age (n = 92) and **b** 0.68 for children ≥5 years of age (n = 87) in the pre-school sample. Source data is in Supplementary Data 2.

retrained the App using the data from the new population, the re-analysis of a subset of images (which were not used for the retraining) with the retrained App showed a relevant improvement in diagnostic accuracy (±2.25 g/dl) and a reduction in bias to 0.25 g/dl. These parameters are similar to the results of the initial validation study by Mannino and colleagues (±2.4 g/d) in a well-controlled setting in the United States[17]. This shows the ability of the App to be adapted to a different context and population and makes it a viable tool in POC settings.

Anemia prevalence in the clinic-based sample is similar to the prevalence in Madhepura district and India according to data from the Indian National Family Health Survey (NFHS-5) of 2019-21. Anemia prevalence was 65.7% among women aged 15–49 years in Madhepura district and 57.0% nationwide compared to 64.3% among women in the clinic-based sample; the respective values for men aged 15–49 were 25.1% nationwide (not available for Madhepura district separately in NFHS-5) and 31.9% in the clinic-based sample[28,29]. In contrast, anemia prevalence among 6–59 months-old children in the pre-school sample was 21.7%, considerably lower than the prevalence according to NFHS-5 for Madhepura district (67.7%) and nationwide (67.1%).

A robust comparison with the performance of other currently available non-invasive POC devices is hampered by highly heterogeneous results stemming from a variety of qualitatively very different studies. This bias of high heterogeneity was also addressed in a representative meta-analysis of 32 studies by Kim and colleagues comparing non-invasive Hb estimation (Masimo tools and NBM-200) with central laboratory measurements[30]. Dimauro and colleagues also pointed out that a comparison of new non-invasive systems is challenging, owing to the considerable differences in the presentation of results and the varying stages of development of the respective approaches[31]. There are also other interesting non-invasive approaches to screen for anemia that do not measure Hb levels, for example measuring zinc protoporphyrin as indicator for iron deficiency, the most common cause of anemia in most populations[32,33]. In general, other non-invasive systems of Hb assessment are also facing challenges in in-field evaluation: the Masimo tools showed very wide LoA (−3.3; 6.9 g/dl) in studies by Gayat and colleagues[34] and by Moore and colleagues (−3.84; 3.89 g/dl)[35]. The NBM-200 by OrSense also demonstrated a great width with LoA ranging from −3.0 to 3.4 g/dl[36].

The present study has several strengths and limitations. The inclusion of both adults and children in the two study samples

allowed for the App to be assessed for a wide age range. Furthermore, the study provides evidence for the App's performance in real-life settings and in a LMIC context. We also considered the technology readiness level (TRL), which is a measure to assess the state of maturity of a new technology and is based on a scale from 1 to 9, with 9 representing the technology with the greatest maturity[37]. The App in its current form is assigned an advanced TRL of 5 or 6, meaning that the technology is validated in a relevant environment. Intensified research including further refinement of the algorithm and studies in real-life settings are essential to raise the App's maturity and the present study has contributed to this process.

Several limitations have to be noted as well. The data at hand do not allow us to study heterogeneity of the App's performance for different Hb values, because for instance only 1.8% of our participants suffered from severe anemia as defined by Hb values below 8 g/dl. The categorization of nails was done manually and therefore a certain degree of subjectivity could not be completely eliminated despite the fact that clear criteria were applied. For ethical as well as logistical reasons, no venous blood sampling was performed among the children of the pre-school sample, so that the HemoCue Hb readings served as reference for comparison with Hb estimates by the App. Although the HemoCue is not the laboratory gold standard, it is commonly used in field research and it is an appropriate reference standard for settings where laboratory capacity is limited or not available.

To conclude, the Sanguina App is an innovative approach towards measuring hemoglobin concentration in LMIC settings. The version used in the present study was by no means a final version, but must be understood as a first stage of development and as a starting point for further adaptation to various populations. Its capacity of learning has been verified by the improved results after retraining the algorithm with fingernail photographs and corresponding Hb values based on venous blood samples. It will not replace the laboratory gold standard and cannot be used in settings where continuous monitoring is necessary. However, it has a wide range of potential applications, at both the population and individual level. It can support public health systems in screening for anemia in settings where access to more sophisticated diagnostic tools is limited and financial resources are scarce. The App could be installed on the smartphones of staff in primary health care facilities in remote regions and can be used after a brief training. This could help accelerate access to next steps in

care. It could also be used as an inexpensive and non-invasive tool for Hb measurements in population-based surveys such as the Demographic and Health Surveys or other research studies that require Hb measurement. It would be a suitable instrument for targeting health intervention and measuring their outcomes. Moreover, the App is in principle also viable for self-screening and self-monitoring. Since it is easy to handle and can be operated by medical laypersons, the App could be used conveniently on-demand from home. Mannino and colleagues reported that it is possible to personalize the algorithm by calibration, which is particularly important for chronically anemic patients who require close monitoring of their Hb concentration[17]. This would be especially helpful for patients in settings where health facilities are far away and difficult to access regularly. The App is a standalone, inexpensive, and rapid tool which has crucial advantages over other POC devices that are currently available[38–40]. The App could play an important role in mitigating the negative consequences of anemia through early detection, especially where health infrastructure is poor and unmet needs are high. Going forward, additional studies and further adaptations are essential to create a credible POC instrument for general use.

## Data availability

All data generated or analysed during this study are included in this published article and its supplementary information files, as Supplementary Data 1 for the clinic-based sample and Supplementary Data 2 for the pre-school sample.

## Code availability

The Stata code used to generate and process the data as described in the manuscript is available in the supplementary information files, particularly Supplementary Data 3 for the analysis of the clinic-based sample and Supplementary Data 4 for the analysis of the pre-school sample.

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

## Acknowledgements

We are very grateful to Robert Mannino for his support. Robert Mannino provided us with access to the App, advised on its use, and made adjustments, so the App could be used on substantially smaller hands of children. He also retrained the algorithm with some of the photos obtained. We would like to thank Rishika Raj and Prerna Swati for their precise and reliable work and kind support during the data collection, Harshpreet Kaur for excellent research assistance, Reetesh Kumar Vimal for his tireless support in the organization of the data collection, the team of the *Madhepura Lab* laboratory for kind cooperation in the data collection, the *Dr. Sitaram Yadav Clinic* in Madhepura for informing their patients about the study and providing the space, and the Anganwadi Workers and Helpers for great support in the data collection in the Anganwadi Centers. We are grateful to the patients of the *Dr. Sitaram Yadav Clinic* in Madhepura and their families as well as the children attending the chosen Anganwadi Centres and their parents for participating in this research.

## Author contributions

V.H., L.B., A.C.W. and S.V. conceived the study, A.K. advised on the implementation of the study. A.K. organized the data collection. V.H. and A.K. managed the data collection. V.H., L.B. and A.C.W. conducted the data analysis, with support from I.D. and S.V. VH wrote the first draft of the manuscript. I.D. and S.V. provided critical feedback to the manuscript structure and contents. All authors contributed to the interpretation of the results and in revising and finalizing of the manuscript and approved the final version.

## Funding

The project received funding of the German Research Foundation via the Research Training Group 1723 at the University of Goettingen. The German Academic Exchange Service supported the field stays within the New Passage to India Program. The App was provided by the developers free of charge and the developers trained our team in the use of the App. The study design was developed completely independently by our team. After data collection, image data was shared with the developers to allow them to retrain the algorithm using this data. Open Access funding enabled and organized by Projekt DEAL.

## Competing interests

The authors declare no competing interests.
