## [Peer Review File · Communications Medicine]

Reviewers' comments:

Reviewer #1 (Remarks to the Author):

The manuscript entitled "Smartphone-based point-of-care method for anemia screening: A 2 validation study from rural Bihar, India" gives analysis of Sanguina Smartphone based app to define its accuracy for measurement of Hb non-invasively. Following are the observations:

1. Repeatability analysis has not given in detail. Please discuss in detail.
2. The accuracy of Sanguina Smartphone App for male patients and female patients has not been shown separately as the ideal values of HB for both are different. Can be discuss in detail in "Discussion" section
3. Instead of nail quality, please discuss the observations about so high limits of agreement by Sanguina Smartphone App.
4. Line 314-318: please mention in detail about the health conditions of patient.
5. Line 328-337: Manual selection of effective areas restricts the app to use in surgical sections, etc. where continuous monitoring is required. Please discuss this in limitations section.

Reviewer #2 (Remarks to the Author):

To
Manuscript Administration
Communications Medicine
4 Crinan Street
London N1 9XW
Manuscript Administration
One New York Plaza, Suite 4500
New York, NY 10004
E-mail: commsmed@nature.com

Dear Sir,

I have attached my comments below following the instructions.

When composing your report, the following questions might assist you in writing an incisive, well-justified review.

What are the major claims of the paper?

The performance of a Sanguina, a mobile-based hemoglobin calculation application, has been evaluated in this study. The authors declared that the App does not meet the accuracy needed for 43 screening purposes. However, the App's performance was improved by retraining the prediction model. Note that the algorithm is not given or shared.

Are they novel and will they be of interest to others in the community and the wider field?

The algorithm's nobility has already been published in the previous study. This study presents how the authors collected some data in a controlled environment from a rural area of India. They collected data from children and adults who are mostly healthy. The author collected data from some subjects who had hemoglobin values less than 8 gm/dL. Most prediction model doesn't work well in the extreme range of hemoglobin values.

If the conclusions are not original, it would be helpful if you could provide relevant references.

The authors trained their algorithm using collected data in India, but this is not an idea of a robust system. Moreover, people will download mobile apps in many parts of the world, so it is not plausible to train the hemoglobin prediction model based on the user's data every time.

Is the work convincing, and if not, what further evidence would be required to strengthen the conclusions?

The actual algorithm of the mobile App is a great invention. But the authors could build a mobile application for the rural user and roll out the App for the real users. Then, the App's performance can be evaluated based on the user's feedback and the accuracy of hemoglobin predictions.

On a more subjective note, do you feel that the paper will influence thinking in the field? Yes, the paper is submitted in the related field.

Please feel free to raise any further questions and concerns about the paper.

Some questions based on authors' statements (with line number) are given below.

Line: 32 – 36: "For 272 mainly adult patients of a private health centre, Hb measured with the App was compared with the gold standard laboratory blood analysis. For a second sample of 179 children attending pre-schools, Hb by App assessment was compared to the results of the HemoCue Hb 301, an invasive point-of-care device serving as the reference for field-based settings."

Reviewer's questions:

1) Why did they compare two different gold standard methods for evaluating the performance? I guess it was impossible to bring the school kids to the clinic.

2) How accurate is the clinically measured hemoglobin value? I am asking because many Indian diagnostic labs do not have NIH-supported standards.

Line 104 -105: Its functionality is based on an extremely color-sensitive analysis algorithm which evaluates the image data of the photos taken from the fingernail beds.

3) How will end-user collect their fingernail data by themselves? I am asking because they don't know when to stop capturing images of finger nailbeds.

Line 113- 116: In the clinic-based sample, a total of 5 photos per participant were captured, either of the right or left hand, in some cases both. In the pre-school sample, between 1 and 4 photos were taken per child, either of the right or left hand, depending on the child's willingness to cooperate and ability to keep the required finger position stable.

4) When will a user understand that the collected data is good enough for the algorithm?

Line 187 - 188: The effectiveness of this retraining of the algorithm was assessed by a re-analysis of validity of a subset of images of the pre-school sample (n=160).

5) How can I be sure that the best image is not selected based on the output generated by the prediction model? So, I am asking, what are the image selection criteria? Do they choose images manually?

Line 230- 235: The scatterplot in Figure 2A reveals a relatively wide scattering along the line of equality. The small Pearson correlation coefficient of $r=0.225$ ($p=0.0002$) confirmed this statistically significant low positive correlation of the two methods. Bland Altman analysis showed 95% LoA of -4.81 to 4.05 g/dl, and a mean difference of -0.38 g/dl, indicating an accuracy of ± 4.43 g/dl (Figure 3A). The average error magnitude was calculated being 1.88 g/dl (Supplemental Table 3A).

Line 244 - 253: Within the pre-school sample, a mean Hb concentration of 11.79 ± 1.11 g/dl was determined for the HemoCue reference method compared to 13.17 ± 1.57 g/dl for App measurements (Supplemental Table 2B). The paired t-test revealed a statistically significant difference in the mean Hb concentration between the two methods ($t=10.61$, $p<0.05$). The Pearson correlation coefficient of $r=0.186$ ($p=0.013$) indicated that there was only a weak positive correlation between App and reference values. The rather wide distribution along the equality line in the associated scatterplot (Figure 2B) also confirmed this observation. Bland Altman technique demonstrated 95% LoA from -2.04 to 4.81 g/dl and a mean difference of 1.39 g/dl (Figure 3B). This yielded an accuracy of ± 3.42 g/dl (Supplemental Table 3B).

6) Is it realistic, and are you confident in deploying this App for hemoglobin prediction with

this performance? Please keep in mind rural users have very minimum experience of smartphone use and mHealth apps. In such a situation, the app users might not know what they are recording using the smartphone camera, and they mistakenly could capture low-quality images of finger nailbeds.

We would also be grateful if you could comment on the appropriateness and validity of any statistical analysis, as well as the ability of a researcher to reproduce the work, given the level of detail provided.

7) Since the Sanguina mobile app is made available on the web, I would highly recommend collecting thousands of different rural users (of a different ages, gender, and hemoglobin ranges)

To increase the transparency and openness of the reviewing process, we do support our reviewers signing their reports to authors if the reviewers feel comfortable doing so.

If, however, you prefer to send an anonymized report we will continue to respect and maintain your anonymity.

Referee reports, whether signed or not, are subsequently shared with the other reviewers.

Md Kamrul Hasan
Assistant Professor of Practice
Department of Computer Science
Vanderbilt University

334 Featheringill Hall
Nashville, TN 37235
kamrul.hasan@Vanderbilt.Edu

Reviewer #3 (Remarks to the Author):

This interesting and important novel study addressing hemoglobin measurements tests a new noninvasive POC technology in the field in a population at-risk. It is an important study in a group with a large number of vegetarians and in a populations with variation in skin color. Note that the app had poor performance initially, but improved remarkably with retraining. It is likely the most important conclusion of the study, but is not addressed in detail in the discussion.

One key aspect of the study not addressed is did the POC app have better performance in those with the lowest Hb vs. higher. Alternatively, did it perform well after picking a threshold of any anemia or moderate-severe anemia vs. those with mild or no anemia? The adult study population had enough to study this question. That is the population at highest risk of complications in resource-poor point-of-care settings. This is the population that clinicians

are most concerned with?

Rationale is well-stated.

Background: Other work on POC testing for ID, the most common cause of anemia in most populations shows some promising results and should likely be referenced in the background.

1. Fullenbach C, et al. Transfusion 2020; Jan;60(1):62-72. Screening for iron deficiency in surgical patients based on noninvasive zinc protoporphyrin measurements. PMID: 31758575 DOI: 10.1111/trf.15577 .

2. Homann C, et al. Pediatric Research 2019; Feb;85(3):349-354. Non-invasive measurement of erythrocyte zinc protoporphyrin in children. PMID: 30655607 DOI: 10.1038/s41390-018-0247-x .

Methods:

1. Was a test of distribution performed for determining which statistics are mean \pm SD, is that the best measure of central tendency, or should it be geometric mean or median.

2. Coefficient of variation or other quality assurance of testing of both the Hemocue and the Aspen Mindray BC-5000 autoanalyzer in practice. Hemocue is reported to have a sensitivity of 0.75 and specificity 1.0 vs. cyanmethemoglobin method, but not sure if this is true in the field.

3. Methodology in the supplementary material should be described in better detail, as these methods were important.

4. Shouldn't ROC methods be included in the main manuscript?

5. Shouldn't SI units g/L for Hb be used for an international journal?

6. The role of the APP developers in this particular study should be described in more detail, specifically did they impact the design or was this study designed independently?

Results:

1. Demographic Characteristics should be described in more detail with groupings such as stunting as malnutrition can result in skin pigmentary disorders, vegetarians, acute illness vs. well, etc. The granular detail is lacking, but a little more detail would be helpful.

2. Other questions about this modality includes does it work with nail fungus? Beta carotemia (at least is it seen in this population based on vegetables high in beta carotene).

3. ROC curves would be helpful in the main manuscript.

4. Going back and forth between the manuscripts figures/tables and the supplement is confusing.

5. The paper goes back and forth between the main manuscript and supplementary materials, making it hard to follow along. This can be addressed by making the supplementary materials more like the main manuscript, including text in the results section and a true discussion section there to be more complete. Then refer back to that discussion.

Thus, major rewriting would be really important, because communication is hindered by this.

Discussion

1. The last statements would be improved by more specifics in the design or testing or improvements needed.

Other comments:

1. Old iPhone OS, potential that the camera quality could improve as subsequent iPhones.
2. Pre-school group had the most variability in the reference Hb Hemocue system, instead of a laboratory-based larger analyzer, but the performance appeared slightly better. What are your thoughts about this.
3. Why are the ROCs and predictive values poorer in women vs. men? This is of concern, since anemia is more common in women vs. men? Why is performance best in kids when the gold standard hemocue has poorer performance? Less nail scarring? other thoughts?
4. More detail about technology readiness level (TRL) would be helpful in the methods or supplementary materials.
5. More detail about what next steps are needed, it is important to note that those with moderate to severe anemia are those most important to diagnose.

Reviewers' comments:

Reviewer #1 (Remarks to the Author):

Thank you for your comments. We revised the manuscript accordingly and provide point-by-point answers below.

The manuscript entitled "Smartphone-based point-of-care method for anemia screening: A 2 validation study from rural Bihar, India" gives analysis of Sanguina Smartphone based app to define its accuracy for measurement of Hb non-invasively. Following are the observations:

1. Repeatability analysis has not given in detail. Please discuss in detail.

Thank you for the comment. We have extended this paragraph and added that reliability was assessed by looking at the intraclass correlation coefficient for a two-way mixed-effects model.

2. The accuracy of Sanguina Smartphone App for male patients and female patients has not been shown separately as the ideal values of HB for both are different. Can be discuss in detail in "Discussion" section. We have now added results of Bland Altman analysis separately for men and women. The ROC analysis, now shifted to the main text, is also shown separately for men and non-pregnant women to take into account their different cut-offs for anemia diagnosis.

3. Instead of nail quality, please discuss the observations about so high limits of agreement by Sanguina Smartphone App.

We have changed the description of nail quality to refer to categories only instead of nail quality and hope that this addresses your concern.

Furthermore, we refrain from speculation about non-observed causes of measurement error. There is no comparison that we can draw from literature.

4. Line 314-318: please mention in detail about the health conditions of patient.

We only have information about the anemia status of the patients. We do not have more detailed health information. For the pre-school sample, we have information on children's anthropometric status which is presented in Supplementary Table 1.

5. Line 328-337: Manual selection of effective areas restricts the app to use in surgical sections, etc. where continuous monitoring is required. Please discuss this in limitations section.

Thank you for your comment. The App is not suitable for continuous monitoring, e.g. during surgery comparable to monitoring oxygen saturation by pulse oximetry, because of the need to take a new picture for each Hb measurement. However, it has a wide range of other potential applications. Most relevant in our opinion is that the App could be installed on the smartphone of staff in primary health care facilities in remote regions and be used after a brief training. This could help accelerate and diagnoses and access to next steps in care. It could also be used as a low-cost tool for Hb measurements in population-based surveys. We have extended the discussion of potential uses in the discussion section and hope that this clarifies our perspective on its potential.

Reviewer #2 (Remarks to the Author):

Dear Sir,

I have attached my comments below following the instructions.

Thank you for your comments. We revised the manuscript accordingly and provide point-by-point answers below.

When composing your report, the following questions might assist you in writing an incisive, well-justified review.

What are the major claims of the paper?

The performance of a Sanguina, a mobile-based hemoglobin calculation application, has been evaluated in this study. The authors declared that the App does not meet the accuracy needed for screening purposes. However, the App's performance was improved by retraining the prediction model. Note that the algorithm is not given or shared.

In the initial study, the App was calibrated to a specific population and demonstrated to work. However, with other populations, adaptation must be done before its validity can be ensured. In our study, the retraining of the algorithm based on the collected data was performed by the developers of the App. Thus, at the time of the study, the algorithm was still under development. The developers were willing to share the App for the purpose of its validation in the framework of this study. However, the research team of this study did not develop the App and is not able to share the algorithm.

Are they novel and will they be of interest to others in the community and the wider field?

The algorithm's nobility has already been published in the previous study. This study presents how the authors collected some data in a controlled environment from a rural area of India. They collected data from children and adults who are mostly healthy. The author collected data from some subjects who had hemoglobin values less than 8 gm/dL. Most prediction model doesn't work well in the extreme range of hemoglobin values.

Although the rate of patients with anemia in the two samples was approximately 45% and 24%, respectively, few patients presented with extremely low Hb concentrations. In the clinic-based sample, only 1.8% suffered from a severe form of anemia, i.e. Hb levels below 8 g/dl (threshold for severe anemia in men, non-pregnant women and children ≥ 5 years of age). Thus, the study has few observations with low Hb values for which the App does not work well. Most are in the range for which the App provides more accurate results.

If the conclusions are not original, it would be helpful if you could provide relevant references.

Since our study has been conducted, there has been one published study in which the performance of the App was assessed and which drew similar conclusions (Young et al. 2021). However, this study was also conducted in a clinic setting rather than in the "field", namely during health screenings of refugees arriving in the US. While that study has a very diverse sample of participants, we still believe that our study provides a unique contribution to the validation of the App. We added the reference to this study to our manuscript.

Young MF, Raines K, Jameel F, et al. Non-invasive hemoglobin measurement devices require refinement to match diagnostic performance with their high level of usability and acceptability. *PLOS ONE*. 2021;16(7):e0254629. doi:10.1371/journal.pone.0254629

The authors trained their algorithm using collected data in India, but this is not an idea of a robust system. Moreover, people will download mobile apps in many parts of the world, so it is not plausible to train the hemoglobin prediction model based on the user's data every time.

Thank you for your comment. We do not argue or want to suggest that the App could be retrained for each user. In the initial study, the App was calibrated to a specific population and demonstrated to work. However, with other populations, adaptation must be done before validity can be ensured. In our study (the 2nd study in general and the 1st in an LMIC context), the App showed poor performance initially, but improved remarkably with retraining, underlining the importance of context specific testing of screening tools. Such phenomena are also present in other methods/applications: Fawzy A, Wu TD, Wang K, et al. Racial and Ethnic Discrepancy in Pulse Oximetry and Delayed Identification of Treatment Eligibility Among Patients With COVID-19. *JAMA Intern Med*. 2022;182(7):730–738.

Is the work convincing, and if not, what further evidence would be required to strengthen the conclusions?

The actual algorithm of the mobile App is a great invention. But the authors could build a mobile application for the rural user and roll out the App for the real users. Then, the App's performance can be evaluated based on the user's feedback and the accuracy of hemoglobin predictions.

The questions you mentioned would certainly be very interesting as well as relevant. However, improving the accuracy of the App by retraining the algorithm is only possible if Hb values from a blood sample are available at the same time. Moreover, we, the authors of this study, are not the developers of the App. We hope that this is now clarified in the manuscript.

On a more subjective note, do you feel that the paper will influence thinking in the field?

Yes, the paper is submitted in the related field.

Great, thanks a lot!

Please feel free to raise any further questions and concerns about the paper.

Some questions based on authors' statements (with line number) are given below.

Line: 32 – 36: "For 272 mainly adult patients of a private health centre, Hb measured with the App was compared with the gold standard laboratory blood analysis. For a second sample of 179 children attending pre-schools, Hb by App assessment was compared to the results of the HemoCue Hb 301, an invasive point-of-care device serving as the reference for field-based settings."

Reviewer's questions:

1) Why did they compare two different gold standard methods for evaluating the performance? I guess it was impossible to bring the school kids to the clinic.

Exactly, for ethical as well as logistical reasons, no venous blood sampling was performed among the children of the pre-school sample, so that the HemoCue Hb readings served as reference.

2) How accurate is the clinically measured hemoglobin value? I am asking because many Indian diagnostic labs do not have NIH-supported standards.

The determination of the reference Hb value was performed by a fully automated hematology analyzer, the "Aspen Mindray BC-5000" autoanalyzer (Mindray, China), in a locally well-established laboratory ("Madhepura Lab"). This laboratory and device were the best possible local standard. The autoanalyzer was used according to the manufacturer's recommendations. We also added a reference presenting the coefficient of variation of HB by this analyzer as 0.58 to 1.06 (Xiang et al. 2015).

Xiang D, Yue J, Lan Y, et al. Evaluation of Mindray BC-5000 hematology analyzer: a new miniature 5-part WBC differential instrument. *International Journal of Laboratory Hematology*. 2015;37(5):597-605. doi:10.1111/ijlh.12370

Line 104 -105: Its functionality is based on an extremely color-sensitive analysis algorithm which evaluates the image data of the photos taken from the fingernail beds.

3) How will end-user collect their fingernail data by themselves? I am asking because they don't know when to stop capturing images of finger nailbeds.

This depends on the application area. In population-based surveys such as the Demographic and Health Surveys, the Hb measurement would be performed by data collectors who would be properly trained in the use of the App in advance. When used in primary health care facilities in remote areas, the App would be used by health care workers who would also receive appropriate training beforehand. The susceptibility to error would probably be greatest when untrained individuals use the App. However, it should also be emphasized that the handling can be learnt by anyone, including medical laypersons, in a very short time. One hand is held in the appropriate position, the other hand can be used to operate the smartphone and take the photo. Step-by-step instructions follow which are designed very intuitively. We added this to the discussion section.

Line 113- 116: In the clinic-based sample, a total of 5 photos per participant were captured, either of the right or left hand, in some cases both. In the pre-school sample, between 1 and 4 photos were taken per child, either of the right or left hand, depending on the child's willingness to cooperate and ability to keep the required finger position stable.

4) When will a user understand that the collected data is good enough for the algorithm?

The repetition of measurements was used only for the purpose of testing repeatability (test-retest reliability). Generally, one photo is sufficient. If the required conditions are met, e.g. correct finger position, no nail polish, the photo usually fulfills all criteria to provide the best possible conditions for the algorithm to calculate the Hb value.

Line 187 - 188: The effectiveness of this retraining of the algorithm was assessed by a re-analysis of validity of a subset of images of the pre-school sample (n=160).

5) How can I be sure that the best image is not selected based on the output generated by the prediction model? So, I am asking, what are the image selection criteria? Do they choose images manually?

In one AWC, there were unforeseen temporary memory constraints on the smartphone, so photos of about half of the children in this centre could not be saved on the device, although their Hb value was recorded. This happened by accident. Thus, not all images of the included children were available for later re-analysis of the retrained algorithm. For this reason, only a subset of n=160 could be considered, but no purposeful selection took place.

Line 230- 235: The scatterplot in Figure 2A reveals a relatively wide scattering along the line of equality. The small Pearson correlation coefficient of $r=0.225$ ($p=0.0002$) confirmed this statistically significant low positive correlation of the two methods. Bland Altman analysis showed 95% LoA of -4.81 to 4.05 g/dl, and a mean difference of -0.38 g/dl, indicating an accuracy of ± 4.43 g/dl (Figure 3A). The average error magnitude was calculated being 1.88 g/dl (Supplemental Table 3A).

Line 244 - 253: Within the pre-school sample, a mean Hb concentration of 11.79 ± 1.11 g/dl was determined for the HemoCue reference method compared to 13.17 ± 1.57 g/dl for App measurements (Supplemental Table 2B). The paired t-test revealed a statistically significant difference in the mean Hb concentration between the two methods ($t=10.61$, $p<0.05$). The Pearson correlation coefficient of $r=0.186$ ($p=0.013$) indicated that there was only a weak positive correlation between App and reference values. The rather wide distribution along the equality line in the associated scatterplot (Figure 2B) also confirmed this observation. Bland Altman technique demonstrated 95% LoA from -2.04 to 4.81 g/dl and a mean difference of 1.39 g/dl (Figure 3B). This yielded an accuracy of ± 3.42 g/dl (Supplemental Table 3B).

6) Is it realistic, and are you confident in deploying this App for hemoglobin prediction with this performance? Please keep in mind rural users have very minimum experience of smartphone use and mHealth apps. In such a situation, the app users might not know what they are recording using the smartphone camera, and they mistakenly could capture low-quality images of finger nailbeds.

After retraining the algorithm and thus adapting it to the circumstances, the App could now be reused in this context. However, we are not confident that the App could be applied in other settings. Other settings would possibly require similar adjustments and retraining as we described in this study. Appropriate training of health care workers/enumerators can ensure that the App is used professionally and that the best possible measurement results are obtained. With background knowledge regarding the interpretation of Hb levels, the App could help accelerate access to next steps in treatment and target/monitor intervention programs. In the setting of our study, frontline workers such as Anganwadi Workers (care takers in the pre-schools) already use smartphones and mobile apps for their daily work.

We would also be grateful if you could comment on the appropriateness and validity of any statistical analysis, as well as the ability of a researcher to reproduce the work, given the level of detail provided.

7) Since the Sanguina mobile app is made available on the web, I would highly recommend collecting thousands of different rural users (of a different ages, gender, and hemoglobin ranges)
This is certainly a great idea, and worth to follow-up in the future. Yet, this is out of scope considering the existing data basis.

Reviewer #3 (Remarks to the Author):

This interesting and important novel study addressing hemoglobin measurements tests a new noninvasive POC technology in the field in a population at-risk. It is an important study in a group with a large number of vegetarians and in a populations with variation in skin color. Note that the app had poor performance initially, but improved remarkably with retraining. It is likely the most important conclusion of the study, but is not addressed in detail in the discussion.

Thank you for your comment. We have now highlighted this in different parts in the manuscript.

One key aspect of the study not addressed is did the POC app have better performance in those with the lowest Hb vs. higher. Alternatively, did it perform well after picking a threshold of any anemia or moderate-severe anemia vs. those with mild or no anemia? The adult study population had enough to study this question. That is the population at highest risk of complications in resource-poor point-of-care settings. This is the population that clinicians are most concerned with?

We have now added results of Bland Altman analysis for subjects with mild or no anemia vs subjects with moderate or severe anemia to the manuscript. The accuracy is very similar and we therefore conclude that the accuracy of the app-based measurement is just as good in high-risk individuals as it is in low-risk groups.

Rationale is well-stated.

Background: Other work on POC testing for ID, the most common cause of anemia in most populations shows some promising results and should likely be referenced in the background.

1. Fullenbach C, et al. Transfusion 2020; Jan;60(1):62-72. Screening for iron deficiency in surgical patients based on noninvasive zinc protoporphyrin measurements. PMID: 31758575 DOI: 10.1111/trf.15577 .
2. Homann C, et al. Pediatric Research 2019; Feb;85(3):349-354. Non-invasive measurement of erythrocyte zinc protoporphyrin in children. PMID: 30655607 DOI: 10.1038/s41390-018-0247-x .

Very interesting approaches, thank you. We have added the references.

Methods:

1. Was a test of distribution performed for determining which statistics are mean \pm SD, is that the best measure of central tendency, or should it be geometric mean or median.

Thank you for this comment. Normal distribution of the variables was first assessed graphically. We have now done Shapiro-Wilk tests to formally test for normal distribution. Descriptive characteristics of the study population are presented as means \pm standard deviations (SD) and ranges for normally distributed continuous variables, as medians and interquartile ranges (IQR) for non-normally distributed variables, and as counts and proportions for categorical variables. As Hb values were not normally distributed according to the Shapiro-Wilk test, we have added IQR in supplementary table 2 which shows the distribution of hemoglobin concentrations.

2. Coefficient of variation or other quality assurance of testing of both the Hemocue and the Aspen Mindray BC-5000 autoanalyzer in practice. Hemocue is reported to have a sensitivity of 0.75 and specificity 1.0 vs. cyanmethemoglobin method, but not sure if this is true in the field.

The Demographic and Health Surveys use the HemoCue as a method of Hb measurement. This system is a simple and relatively low-cost technique that has been accepted as a standard method for hemoglobin measurement by the International Committee for Standardization in Hematology (<https://dhsprogram.com/pubs/pdf/OD22/OD22.pdf>).

In our study in a subgroup of the clinic-based sample (n=106), the HemoCue was tested against the laboratory reference standard prior to data collection in pre-schools. The calculated accuracy of ± 1.26 g/dl in Bland Altman

analysis is consistent with findings from other studies (Hiscock et al. 2015) and considered sufficient for requirements in a field-based context.

For the Aspen Mindray BC-5000 autoanalyzer, we now added a reference presenting the coefficient of variation of Hb as 0.58 – 1.06 (Xiang et al. 2015).

Hiscock R, Kumar D, Simmons SW. Systematic review and meta-analysis of method comparison studies of Masimo pulse co-oximeters (Radical-7 or Pronto-7) and HemoCue(R) absorption spectrometers (B-Hemoglobin or 201+) with laboratory haemoglobin estimation. *Anaesth Intensive Care*. 2015;43(3):341-50.

Xiang D, Yue J, Lan Y, et al. Evaluation of Mindray BC-5000 hematology analyzer: a new miniature 5-part WBC differential instrument. *International Journal of Laboratory Hematology*. 2015;37(5):597-605.
doi:10.1111/ijlh.12370

3. Methodology in the supplementary material should be described in better detail, as these methods were important.

We have now shifted as much as possible to the main manuscript while still adhering to the journal guidelines of a maximum of 10 items (figures and tables) and 5000 words in the main manuscript. The supplementary material now only contains the descriptive characteristics, descriptives related to repeatability, and the description of the relationship between nail categories and validity. All methods are described in the main text.

4. Shouldn't ROC methods be included in the main manuscript?

We have now included the ROC analysis including ROC curves in the main section.

5. Shouldn't SI units g/L for Hb be used for an international journal?

g/dl is commonly used in the context of Hb measurement. To ensure better comparability with the results of the initial study of Mannino and colleagues, we have chosen g/dl instead of g/l. Of course we will adhere to the journal policy and change this to g/l if requested by the editors.

6. The role of the APP developers in this particular study should be described in more detail, specifically did they impact the design or was this study designed independently?

The App was provided by the developers free of charge. The study design was developed completely independent from the app developers by our research team. After data collection, an exchange of data was done, used by the developers for the retraining process of the algorithm. We have added this information in the manuscript.

Results:

1. Demographic Characteristics should be described in more detail with groupings such as stunting as malnutrition can result in skin pigmentary disorders, vegetarians, acute illness vs. well, etc. The granular detail is lacking, but a little more detail would be helpful.

In the clinic-based sample, age, sex and diet requirements (vegetarian or non-vegetarian) were recorded and are shown in table 1. Demographic characteristics, including age and sex, as well as anthropometric measurements of height, weight and mid-upper arm circumference (MUAC) were obtained from children of the pre-school sample. A calculation of HAZ, WAZ, WHZ, BAZ and nutritional status (stunting, underweight, wasting, thinness, acute malnutrition and CIAF) was performed and is shown in supplementary table 1. We added detail to the paragraph describing children's characteristics. Unfortunately, we do not have further information on the health status of the patients.

2. Other questions about this modality includes does it work with nail fungus? Beta carotemia (at least is it seen in this population based on vegetables high in beta carotene).

If the nail fungus affects only a part of the nail surface, this part can be excluded by selecting the box (ROI) outside of it and is thus not included in analysis.

3. ROC curves would be helpful in the main manuscript.

The ROC analysis is now part of the main manuscript.

4. Going back and forth between the manuscripts figures/tables and the supplement is confusing.

We have now shifted as much as possible to the main manuscript while still adhering to the journal guidelines of a maximum of 10 items (figures and tables) and 5000 words in the main manuscript. The supplementary material now only contains the descriptive characteristics, descriptives related to repeatability, and the description of the relationship between nail categories and validity. All methods are described in the main text. We hope that this addresses the concern.

5. The paper goes back and forth between the main manuscript and supplementary materials, making it hard to follow along. This can be addressed by making the supplementary materials more like the main manuscript, including text in the results section and a true discussion section there to be more complete. Then refer back to that discussion.

We have now shifted as much as possible to the main manuscript while still adhering to the journal guidelines of a maximum of 10 items (figures and tables) and 5000 words in the main manuscript. The supplementary material now only contains the descriptive characteristics, descriptives related to repeatability, and the description of the relationship between nail categories and validity. All methods are described in the main text. We hope that this addresses the concern.

Thus, major rewriting would be really important, because communication is hindered by this.

Discussion

1. The last statements would be improved by more specifics in the design or testing or improvements needed.
We have added more information in the manuscript.

Other comments:

1. Old iPhone OS, potential that the camera quality could improve as subsequent iPhones.

The iPhone 5s was used for the study, as the App was developed for it (this is now mentioned in the manuscript). We therefore wanted to make sure that our data compared well with that of the initial validation study by Mannino et al. Since then, the App has been further developed and can be used on many different devices. We did not assess a relation between the quality of results and the camera quality.

2. Pre-school group had the most variability in the reference Hb Hemocue system, instead of a laboratory-based larger analyzer, but the performance appeared slightly better. What are your thoughts about this.

The two groups are somewhat difficult to compare as the reference method was different for the two groups. We therefore do not want to hypothesize about potential reasons but want to emphasize the general poor initial performance of the App and the subsequent improvement after retraining. The latter could then only be shown for the pre-school sample as the clinic-based sample was used for the retraining.

3. Why are the ROCs and predictive values poorer in women vs. men? This is of concern, since anemia is more common in women vs. men? Why is performance best in kids when the gold standard hemocue has poorer performance? Less nail scarring? other thoughts?

This is a valid point. However, we can only speculate about this. As for the poorer performance among women compared to men, this could potentially be explained by the greater variability in Hb among women due to their menstrual cycle. We added this potential explanation with a reference (Murphy 2015) in the discussion section. As for the better performance in kids: less nail scarring could be one factor explaining the better performance. However, we also note that while accuracy is better among the pre-school sample, the bias is larger.

Murphy WG. The sex difference in haemoglobin levels in adults — Mechanisms, causes, and consequences. *Blood Reviews*. 2014;28(2):41-47. doi:10.1016/j.blre.2013.12.003

4. More detail about technology readiness level (TRL) would be helpful in the methods or supplementary materials.

We have added an explanation in the manuscript.

5. More detail about what next steps are needed, it is important to note that those with moderate to severe anemia are those most important to diagnose.

We have added more detailed information in the conclusion.

REVIEWERS' COMMENTS:

Reviewer #3 (Remarks to the Author):

The authors responded to reviewer suggestions.

Reviewer #2 confirmed confidentially to editors that issues were adequately addressed.

Reviewer #3 confirmed confidentially to editors that issues raised by Reviewer #1 had been adequately addressed.

Reviewers' comments:

Reviewer #3 (Remarks to the Author):

The authors responded to reviewer suggestions.

Thank you for your renewed revision of the manuscript and approval of our responses to your suggestions.